# Characterization of Beverage Viscosity Based on the International Dysphagia Diet Standardisation Initiative and Its Correspondence to the Japanese Dysphagia Diet 2021

**DOI:** 10.3390/nu17061051

**Published:** 2025-03-17

**Authors:** Mari Nakao-Kato, Aya Takahashi, Jin Magara

**Affiliations:** 1Division of Health and Nutrition, Department of Home Economics, Tohoku Seikatsu Bunka University, 1-18-2, Nijinooka, Izumi-ku, Sendai 981-8585, Miyagi-ken, Japan; aya.takahashi@mishima.ac.jp; 2Department of Otolaryngology-Head and Neck Surgery, Graduate School of Medicine, Tohoku University, 1-1, Seiryomachi, Aoba-ku, Sendai 980-8574, Miyagi-ken, Japan; 3Division of Dysphagia Rehabilitation, Niigata University Graduate School of Medical and Dental Sciences, 2bancho-5274 Gakkocho-dori, Chuo-ku, Niigata 951-8514, Niigata-ken, Japan; jin-m@dent.niigata-u.ac.jp

**Keywords:** dysphagia, deglutition disorder, International Dysphagia Diet Standardisation Initiative, Japanese Dysphagia Diet, thickened liquid, diet standardization

## Abstract

**Background/Objective:** The International Dysphagia Diet Standardisation Initiative (IDDSI) and the Japanese Dysphagia Diet 2021 (JDD2021) are prominent systems that classify thickened beverages for dysphagia management. We herein aim to establish a correspondence between these systems through systematic viscosity measurements. **Methods:** We analyzed 49 thickened beverage samples using an E-type viscometer, IDDSI flow test, and JDD syringe test. **Results:** Receiver operating characteristic analysis revealed the following viscosity cutoffs for IDDSI levels: 0–1 at 72.0 mPa·s (area under the curve [AUC] 0.94), 1–2 at 112.0 mPa·s (AUC 0.95), and 2–3 at 303.0 mPa·s (AUC 0.97). Multiple regression analysis revealed that beverage characteristics, including fat, sodium content, and settling time, significantly influenced viscosity (R^2^ = 0.803). The findings established that IDDSI Level 0 corresponds to a thinner viscosity than JDD Stage 1 and Stage 1 (0–72 mPa·s), Level 1 to Stage 1 (72–112 mPa·s), Level 2 to Stages 1–2 (112–303 mPa·s), and Level 3 to Stage 3 (>303 mPa·s). Moreover, the measurement methods had high correlations (r = 0.83–0.93, *p* < 0.001). **Conclusions:** The comprehensive map between IDDSI and JDD2021 classifications developed from the results of this study enables healthcare providers to translate between the classification systems, improving dysphagia management internationally while supporting evidence-based care and global research.

## 1. Introduction

Dysphagia, i.e., difficulty in swallowing food and liquids, is a critical global health challenge affecting diverse patient populations [1,2,3,4,5,6]. According to a recent epidemiological study, dysphagia affects approximately 5.9 million people globally [7]. In the United States, one in six adults reported experiencing difficulty swallowing. However, half of these individuals have not discussed their symptoms with a clinician [8]. The prevalence of this condition varies markedly across demographics, particularly among older adults. In Japan, the prevalence of dysphagia among adults aged 50–79 years living in cities is 17.8% for women and 21.9% for men [9]. In a systematic review and meta-analysis, the pooled prevalence of dysphagia in residential aged care facilities was found to be 56.11% [10]. In a study of acute care hospitals in Spain, the prevalence in hospitalized adults over 80 years old was 82.4% [11].

In the management of dysphagia, to ensure safe swallowing and adequate nutrition, careful modification of food and liquid composition is necessary. This clinical need has led to the development of the following standardized classification systems for texture-modified diet (TMD) and thickened drinks: the International Dysphagia Diet Standardisation Initiative (IDDSI) [12] and the Japanese Dysphagia Diet 2021 (JDD2021) [13] established by the Japanese Society for Dysphagia Rehabilitation. These systems have similar clinical objectives but employ different classification and measurement approaches.

The IDDSI framework, established in 2016 [12] and subsequently revised in 2019 [14], comprehensively attempts to create a globally standardized approach for the dietary classification of dysphagia. This system has been officially adopted in many countries, with reference groups established in 128 countries and the guidelines translated into 50 languages [15]. Especially in Asia, several countries, such as Singapore, China, Hong Kong, Malaysia, and Thailand, have officially adopted the IDDSI as a national standard, with widespread use in Taiwan [16].

However, Japan has developed its own classification system, recently updated as the JDD2021 [13]. This system is built on the previous Japanese Dysphagia Diet 2013 (JDD2013) [17]; however, it incorporates elements of the IDDSI framework. In particular, the JDD2021 uses a syringe test, which is similar in principle to the IDDSI flow test, for the assessment of liquids. Moreover, the JDD2021 includes special considerations for Japanese dietary habits and clinical traditions while aligning with international standards [13].

The fundamental difference between the IDDSI and JDD2021 is their approach to measurement and classification. The JDD2021 defines a viscosity range for each level of liquid thickening intensity and provides numerical ranges for classification [13], whereas the IDDSI does not specify viscosity values [12]. This creates challenges in establishing a direct correspondence between the systems, with significant implications for clinical practice, research methods, and international standardization efforts.

The need for a clear correspondence between these systems extends beyond academic interest. Medical care workers are increasingly working in international settings, such as practicing in different countries, collaborating with international teams, or treating patients from diverse backgrounds, particularly in Asia, where there is a need to understand the compatibility between the two systems. The increasing number of health or care suppliers from Asia in Japanese medical care facilities makes establishing a clear relationship between the IDDSI and JDD2021 classification systems crucial. Moreover, manufacturers of dysphagia-specific products need to understand these relationships to ensure proper labeling and distribution in different markets.

Recent studies [18,19] have demonstrated the utility of the IDDSI criteria in improving the safety and nutritional management of patients with dysphagia. Moreover, studies examining the impact of standardized beverage assessment and TMD in elderly care facilities have shown promising results in reducing the aspiration risk and improving on patient outcomes [20,21]. Furthermore, the application of the IDDSI criteria to various dietary cultures has been examined, with a particular focus on traditional Asian foods and preparation methods [22,23].

This study aimed to address the critical need for a correspondence between the IDDSI and JDD2021 classification systems, particularly focusing on thickened beverages. Unlike previous studies, we first established accurate viscosity parameters for the IDDSI levels using systematic measurements and analysis, followed by mapping these parameters to JDD2021’s viscosity ranges. This method has several advantages over direct system comparisons, potentially yielding more reliable and clinically applicable results.

The specific objectives of this study were to characterize the viscosity range corresponding to each IDDSI level using multiple measurement techniques, establish a statistical relationship between IDDSI flow test results and absolute viscosity measurements, develop a comprehensive map between the IDDSI and JDD2021 classifications based on empirical data, and develop a comprehensive map between the IDDSI and JDD2021 classifications.

This study contributes to the broader goal of the international standardization of dysphagia care while acknowledging and addressing regional differences in clinical practice. The results of this study may be particularly important to healthcare providers working across different systems, researchers conducting international studies, and manufacturers developing products for multiple markets. Moreover, the results will provide a basis for future research on the practical implementation of standardized TMD classifications in diverse healthcare settings.

## 2. Materials and Methods

To clarify the correspondence between the two systems, IDDSI ver. 2.0 and JDD2021 standards for thickened beverages were used.

### 2.1. Sample Creation

Ten commercial beverages (water; green tea [Oi Ocha^TM^ by Itoen Co., Ltd., Tokyo, Japan]; 100% orange juice [Pom Juice^TM^ by Ehime Beverage Inc., Matsuyama, Japan]; milk [Meiji Oishii Gyunyu^TM^ by Meiji Co., Ltd., Tokyo, Japan]; coffee [Boss^TM^ Black Coffee, unsweetened by Suntry, Tokyo, Japan]; black tea [Afternoon Tea^TM^, straight, unsweetened by Kirin Beverage Co., Ltd., Tokyo, Japan]; oolong tea [by Suntry]; sports drinks [Pocari Sweat^TM^ by Otsuka Pharmaceutical Co., Ltd., Tokyo, Japan]; tomato juice [by Kagome Co., Ltd., Tokyo, Japan]; and a lactic acid beverage [Yakult^TM^ by Yakult Honsha Co., Ltd., Tokyo, Japan]) were used, and thickened beverages of five different viscosities were prepared by adding thickening agents. The target viscosities of the beverages were 30, 50, 100, 200, and 400 mPa·s. The quantities of thickening agent used are listed in Table 1. The thickening agent was a xanthan-based third-generation thickener (Neo High Toromir III by Food Care Cooperation, Hashimoto, Japan). The temperature of the beverage was adjusted to 20 °C ± 2 °C at the time of measurement (measured using a central thermometer). The thickener was mixed with sample beverages for 30 s. For some beverages in which the thickener was recommended to be mixed twice by the thickener producing company, the sample beverages were mixed with the thickener for 30 s, allowed to settle for the required static time, and again mixed with the thickener for 30 s [24]. The preparation methods are shown in Table 1. For beverages (water and tea) where the thickening agent manufacturer had published information on mixing and settling, the setting adhered to the manufacturer’s instructions. For beverages where the manufacturer of the thickening agent did not publish any instructions, the setting decisions were made by the research team based on a previous study [25], considering the characteristics of the solution, such as salt content, pH, and lipid and protein content.

### 2.2. Measuring Instruments and Conditions

The following instruments were used to measure the viscosity and residual volume. Each sample was measured twice.

Viscometry: The viscosity of each sample was measured using an E-type simple viscometer (JOVI^TM^, Neutri Corporation, Mie, Japan). The shear rate was 50 s^−1^, and the maximum viscosity of measurement was 500 mPa·s. For the measurement, two viscometers were prepared, each was calibrated, and two inspectors measured one liquid sample simultaneously.IDDSI flow test: A total of 10 mL of the sample liquid was injected into a funnel officially approved by the IDDSI and commercially available. Then, the funnel tip was opened, the liquid was dropped for 10 s, and the funnel tip was closed to measure the residual volume (mL) [26]. The liquid was then classified based on IDDSI levels 0–3. Two inspectors simultaneously measured one liquid sample using two funnels.JDD syringe test: Using a Terumo 10 mL syringe, which was officially adopted in JDD2021, 10 mL of the sample liquid was injected, the syringe tip was opened, the liquid was dropped for 10 s, and the syringe tip was closed to measure the residual volume (mL) [27]. The liquid was then classified according to the JDD2021 classification [13,23]. In JDD2021, the thickness of the liquids is classified into three categories, stage 1 (mildly thick), stage 2 (moderately thick), and stage 3 (extremely thick). Liquids thinner than stage 1 are considered inappropriate for patients with dysphagia. Two examiners measured each liquid sample simultaneously using two syringes.

### 2.3. Measurement Process

To characterize IDDSI levels in terms of viscosity and establish correspondence with the JDD2021 standards, a systematic evaluation of thickened liquids was conducted as follows:The thickened beverages were mixed to ensure uniformity and adjusted according to the method indicated in Section 2.2.The prepared samples were measured twice using two separate viscometers, and the viscosity was recorded in mPa·s.The IDDSI flow test and JDD syringe test were performed using samples prepared under the same conditions as the viscosity measurement to avoid time differences and temperature changes from preparation to measurement, and the residual volume (mL) of each was recorded. Each sample was measured twice.The average value of the results (two measurements) was calculated for each sample.

### 2.4. Statistical Analysis

We employed several statistical approaches to characterize the viscosity range of the IDDSI levels:Descriptive statistics: Mean viscosities were calculated for each beverage sample (Table 2). The maximum, minimum, and median viscosities were recorded. All samples were classified using the JDD syringe test and the IDDSI flow test. The distribution of viscosity and residual volume are shown in the descriptive statistics. Scatter plots show the distributions of the viscosity measurement and the residual volume of the JDD syringe test using the Terumo syringe (Figure 1), the residual volume of the IDDSI flow test using the IDDSI funnel (Figure 2), and the distribution of the IDDSI funnel residue and the Terumo syringe residue (Figure 3). The matching ratios between the results from the viscometer and the JDD classification obtained from the value in the Terumo syringe test were calculated for each JDD stage.

2.Correlation analysis: Using Spearman’s rank correlation analysis, we evaluated the correlation between residual volumes and viscosity measurements obtained through the IDDSI flow test and the JDD syringe test. The correlation between the residual volume of the IDDSI funnel and that of the Terumo syringe was also evaluated and correlation coefficients were calculated.3.Receiver operating characteristic (ROC) analysis: This analysis was performed to determine the viscosity cutoff value for each IDDSI level.4.Multiple regression analysis: This analysis was performed using viscosity measurements (log-transformed) as the dependent variable and IDDSI funnel residue (residual volume), fat content of the beverage, sodium content, pH, and static time for settling as independent variables to determine the extent to which the beverages’ nature influenced the determination of viscosity and IDDSI funnel residue [22,23]. The static time was used as a ranking variable, with 1 being 5 min of static time, 2 being 10 min of static time, and 0 being no settling time.

The following pieces of software were used for statistical analysis of the data: SPSS ver. 30 (IBM Corporation) for descriptive analysis, correlation analysis, and multiple regression analysis; Graphpad Prism ver. 10.2.2 for scatterplot creation; and R ver. 4.2.3 for ROC analysis, and the confidence interval for sensitivity and specificity of ROC analysis.

## 3. Results

### 3.1. Descriptive Statistics

The measurement results of the 49 samples are listed in Table 2. Viscosity is shown as the average of two measurements. The viscosity of the 49 samples ranged from 19 to 492 mPa·s, with a median of 100 mPa·s. When the 49 samples were classified using the JDD syringe test, the distribution of the beverage samples was 19–170.5 mPa·s for a thinner viscosity than stage 1, 58–270 mPa·s for stage 1, 192–479 mPa·s for stage 2, and 392–492 mPa·s for stage 3. When classified using the IDDSI funnel flow test, the distribution of the beverage samples was 19–170 mPa·s for level 0, 74.5–280 mPa·s for level 1, 124–479 mPa·s for level 2, and 400–492 mPa·s for level 3.

### 3.2. Correlation Analysis

The relationship between the viscosity and residual volume is shown in Figure 1 and Figure 2. The viscosity measured using the E-type viscometer and the residual volume of the Terumo syringe showed a high correlation (r = 0.83, *p* < 0.001, Spearman’s rank correlation analysis) (Figure 1). However, when the value obtained using the Terumo syringe test was classified according to the JDD2021 classification, the following results were obtained for the samples: 92% of the samples classified as stage 1 or less (residual volume of 0 to 2.2 mL or less) in the Terumo syringe test fell into the 0–50-mPa·s range in the viscosity measurement and 46% of the samples classified as stage 1 (residual volume of 2.2 to 7.0 mL) in the Terumo syringe test fell into the 50–150-mPa·s range in the viscosity measurement. Only 23% of the samples classified as stage 2 (7.0 to 9.5 mL residual volume) in the Terumo syringe test fell in the 150–300-mPa·s range in the viscosity measurement, and only 40% of the samples classified as stage 3 (9.5 to 10 mL residual volume) in the Terumo syringe test fell in the 300–500-mPa·s range in the viscosity measurement.

The accuracy of the classification method varied across the viscosity range. In the present samples, the agreement between the residual volume of the Terumo syringe and the thickening stage of viscosity determined using the E-type simple viscometer was relatively good for thinner viscosity than stage 1 and stage 1 beverages, and poor for stage 2 and above beverages. To confirm the reliability of the residual volume from the Terumo syringe, the agreement between the thickening stages of the JDD determined from the E-type viscometer data and that determined from the Terumo syringe was assessed using an intraclass correlation coefficient (ICC 3.1). The intraclass correlation coefficient was 0.855 (95% CI 0.757–0.916) and Cronbach’s alpha was 0.92 (*p* < 0.001).

Similarly, the viscosity measured using the E-type viscometer and the residual volume of the IDDSI funnel showed a high correlation (r = 0.83, *p* < 0.001, Spearman’s rank correlation analysis) (Figure 2). The residual volumes of the Terumo syringe and IDDSI funnel showed an even higher correlation (r = 0.93, *p* < 0.001, Spearman’s rank correlation analysis) (Figure 3). The distribution of the 49 samples on JDD2021 and IDDSI ver2.0 is shown in Table 3.

### 3.3. ROC Analysis

ROC analysis was performed to determine the cutoff for each level of DDSI. The JDD2021 specifies the following cutoff for each thickening stage, 50–150 mPa·s for stage 1, 150–300 mPa·s for stage 2, and 300–500 mPa·s for stage 3 [13]. The cutoff values for different levels of IDDSI beverages have not been clarified by viscosity. In this study, ROC analysis was performed using the viscosity and the data of funnel residue from 49 samples to determine the cutoff with maximum sensitivity and specificity (Figure 4). The cutoff was 72.0 mPa·s (area under the curve [AUC] 0.94, sensitivity 0.96, 95% CI 0.89–1.0, specificity 0.82, 95% CI 0.64–0.95) between IDDSI levels 0 and 1, 112.0 mPa·s (AUC 0.95, sensitivity 1.0, 95% CI 1.0–1.0, specificity 0.77, 95% CI 0.64–0.90) between levels 1 and 2, and 303.0 mPa·s (AUC 0.97, sensitivity 1.0, 95% CI 1.0–1.0, specificity 0.93, 95% CI 0.83–1.0) between levels 2 and 3.

### 3.4. Multiple Regression Analysis

To determine the extent to which beverage properties influence the determination of viscosity and residual volume in the IDDSI funnel, a multiple regression analysis was performed using the raw data of two measurements for each sample (n = 97), excluding one sample from milk and two samples from oolong tea (Appendix A). These were excluded because they exceeded the measurement limitations. The results showed that physical measurements and beverage characteristics had a significant influence on viscosity. The multiple regression analysis performed resulted in the following regression equation:(1)$$\begin{array}{c} log\left(Viscosity+1\right)\\ \;=3.88+0.836\cdot log\left(FunnelResidue+1\right)-0.280\\ \;\cdot FatContent-0.007\cdot SodiumContent+0.432\\ \;\cdot StaticTime, \end{array}$$

Multiple regression analysis revealed that this model (1) explained 80.3% of the viscosity, with a coefficient of determination (R^2^) of 0.803. For each coefficient, the funnel residue, fat content, sodium content, and static time were statistically significant. The partial coefficients of determination were 69.9% for log funnel residue, 7.5% for fat content, 1.1% for sodium content, and 7.0% for static time.

A summary of the results and the correspondence between JDD2021 and JDDSI ver. 2.0 for beverages are shown in Figure 5. The comparative analysis of the IDDSI and JDD2021 classifications revealed a systematic relationship but not a consistent one-to-one correspondence. For the JDD cutoff values of 50, 150, and 300 mPa·s and the IDDSI ver. 2.0 cutoff values of 72, 112, and 303 mPa·s, IDDSI level 0 corresponded to a thinner viscosity than JDD stage 1 and stage 1, level 1 to JDD stage 1, level 2 to JDD stage 1 and 2, and level 3 to almost JDD stage 3 beverages.

## 4. Discussion

To the best of our knowledge, this is the first study to comprehensively characterize the relationship between viscosity ranges corresponding to IDDSI levels and the JDD2021 standard. Although the viscosity ranges between the IDDSI levels are clearly distinguishable, our results revealed significant variability in their correspondence with the JDD2021 classification. These findings raise several challenges for clinical practice, research methodology, and international standardization efforts in dysphagia management.

### 4.1. IDDSI Level Viscosity Characteristics

The IDDSI framework intentionally avoids specifying viscosity ranges to maintain flexibility across different measurement approaches and cultural backgrounds [14]. However, our ROC analysis demonstrated that transition points between the IDDSI levels have statistically high reliability. A particularly strong discrimination capability was observed between levels 2 and 3 (AUC: 0.97, sensitivity: 1.0, specificity: 0.93). These results indicated that despite the lack of explicit viscosity criteria in the IDDSI, objective thresholds based on actual measurements exist.

### 4.2. Correspondence Between the JDD2021 and IDDSI

Our study clarified the relationship between the JDD2021 and IDDSI based on viscosity (mPa·s) and IDDSI funnel residue (mL), which is as follows:IDDSI level 0 (0–72 mPa·s, funnel residue 0.0–1.0 mL): Corresponds to liquids thinner than JDD stage 1 and some stage 1 liquids.IDDSI level 1 (72–112 mPa·s, funnel residue 1.0–4.0 mL): Corresponds to JDD stage 1 (mildly thick).IDDSI level 2 (112–303 mPa·s, funnel residue 4.0–8.0 mL): Spans across JDD stage 1 (mildly thick) and portions of stage 2 (moderately thick).IDDSI level 3 (>303 mPa·s, funnel residue 8.0–10.0 mL): Closely corresponds to JDD stage 3 (extremely thick).

Contrary to expectations, the relationship between both classification systems was not a simple one-to-one correspondence but showed complex overlapping patterns. High-viscosity liquids (IDDSI level 3 and JDD stage 3) showed strong agreement, while intermediate to low-viscosity regions exhibited significant overlap between categories. This finding has important clinical implications, particularly in healthcare environments where both systems are used concurrently. The strong agreement for high-viscosity liquids suggests that system conversion is reliable for patients requiring thicker compositions, while handling low- to medium-viscosity liquids requires special caution.

### 4.3. Practical Impact on Dysphagia Management in Clinics or Facilities

Our findings have the following specific implications for the practical management of dysphagia in clinics or facilities:Refinement of clinical assessment and prescription: The clarified viscosity correspondence table enables clinicians to accurately convert patient assessment results and prescription contents between different classification systems. This facilitates communication during international hospital transfers or among multinational staff familiar with different systems. Notably for patients with severe dysphagia (requiring high-viscosity thickeners), consistent and safe care can be provided across systems.Individualized approaches based on beverage characteristics: The discovery that beverage characteristics such as fat content and static time influence measurement values demonstrates the need for individualized thickener adjustments in clinical settings. Even with identical thickener concentrations, viscosity is dependent on the beverage type, providing scientific evidence for the individual adjustment of patients’ preferred beverages. This contributes to optimizing patient meal satisfaction and nutritional intake.Enhanced risk assessment: The finding that flow measurement accuracy varies by viscosity range necessitates a more refined aspiration risk assessment, particularly in the intermediate viscosity range (IDDSI level 2 or JDD stages 1–2). Clinicians can exercise more careful interpretation of measurements in this range and, when necessary, combine multiple measurement methods to achieve a more accurate assessment of risk and to aid in planning care.Streamlined interfacility and international collaboration: The established conversion framework facilitates information sharing and continuity of care between facilities using different classification systems. This provides a practical tool for ensuring continuity of safe care, especially in international medical institutions or in facilities with patients and staff of several different nationalities.

### 4.4. Potential for Improvement in Standardization Processes

Our findings suggest the following improvements in standardization processes for dysphagia management:Need for an integrated approach for viscosity measurement: Current standardization processes do not adequately consider fundamental differences between systems. While the IDDSI does not specify viscosity values, the JDD2021 defines specific viscosity ranges. Future standardization would benefit from an integrated approach combining flow measurements and absolute viscosity values to enhance scientific precision while respecting different cultural backgrounds and clinical traditions.Introduction of correction factors based on beverage characteristics: Results of multiple regression analysis suggest the value of introducing correction factors based on beverage characteristics (fat content, sodium content, static time, etc.) into standardization processes. This would improve measurement consistency across different beverage types, enabling more reliable classification.Development of evaluation methods optimized for specific viscosity ranges: Our research demonstrated that measurement accuracy varies for each viscosity range. Future standardization requires development and validation of assessment methods specialized for each viscosity range. New measurement protocols and equipment to improve reliability in the moderate viscosity range are particularly desirable.Flexible framework reflecting cultural diversity and clinical realities: International standardization must respect differences in cultural eating habits and medical resources. Future improvements should focus on building flexible frameworks that allow for regional adaptations while maintaining scientific consistency. For example, supplementary guidelines should be developed for region-specific beverages and foods.Establishment of verification systems based on clinical outcomes: The final evaluation of standardization processes should be based on clinical outcomes. An international verification system to systematically assess how different classification systems affect actual aspiration rates, pneumonia incidence, nutritional status, and patient satisfaction is desirable.

### 4.5. Reliability and Limitations of Measurement Methods

The high correlation between measurement methods (r = 0.83–0.93) supports the overall reliability of the IDDSI and JDD2021 test methods. However, their accuracy varies significantly by viscosity range. In particular, with Terumo syringe classification, marked differences were observed between low-viscosity liquids (92% accuracy) and high-viscosity liquids (23–46% accuracy). This difference is likely attributable to physical factors, such as the non-Newtonian flow characteristics of high-viscosity liquids affecting measurement equipment performance. This finding provides important implications for clinical evaluations (videofluoroscopy or endoscopic swallowing assessment) and quality control procedures in medical settings.

### 4.6. Research Limitations and Future Directions

This study had several limitations. First, the relatively small sample size of 49 samples may affect the generalizability of results. Second, the accuracy of flow tests varied significantly by viscosity level, presenting challenges for the precise measurement of high-viscosity liquids [28]. Third, the nonlinear relationship between viscosity and flow test results may affect accuracy in specific ranges [29,30]. Fourth, the shear-thinning behavior specific to xanthan-based thickeners may affect measurement consistency [26] and potentially limit the generalizability of findings to other thickener types.

Considering these constraints, future research should verify relationships across larger sample sizes and a wider range of beverages and thickeners commonly used in clinical settings. In addition, investigating temperature effects and stability of thickened liquids over time can provide important insights for practical applications. Furthermore, research on clinical outcomes associated with different classification systems can contribute to establishing optimal conversion methods between classification systems and improving patient care.

### 4.7. Additional Research Needed for International Standardization

To achieve true international standardization of thickened beverages and ensure reliable integration of different classification systems, the following additional research is essential:Multinational collaborative validation studies: Large-scale collaborative research involving research institutions representing multiple countries and regions is necessary. This would identify regional differences in measurement methods and interpretation and allow for the development of universally applicable validation protocols. The cross-validation of measurements between regions with different classification systems and clinical standards, such as Japan, Europe, North America, and Australia, is of particular importance.Systematic comparison of diverse thickener types: While this study focused on xanthan-based thickeners, systematic comparisons of the flow characteristics of thickener types used worldwide, including starch-based and guar gum-based products, is needed. This would enable the development of algorithms to predict and correct for variations in measurements due to thickener type.Patient-centered outcome research: Long-term prospective studies evaluating how different classification systems and viscosity levels affect real-world patient outcomes (aspiration pneumonia rates, nutritional status, QOL, patient satisfaction, etc.) are needed. It is especially important to clarify how subtle viscosity differences near system conversion points affect clinical results.Analysis of cultural and dietary habit influences: Research investigating how regional food cultures and beverage preferences affect viscosity perception and optimal viscosity levels is necessary. It has been suggested that the effect of adding a thickening agent to alcoholic and other flavored beverages could be evaluated. This would enable the development of standardization approaches that maintain scientific consistency while respecting cultural diversity.

## 5. Conclusions

This study fully achieved its intended objective of characterizing the relationship between viscosity ranges corresponding to IDDSI levels and the JDD2021 standard. Using multiple measurement techniques, the viscosity characteristics of each IDDSI level were clearly defined and statistical relationships between IDDSI flow test results and absolute viscosity measurements were established. Our research will contribute to the development of international clinical practices, research methods, and standardization efforts in dysphagia management.

## Figures and Tables

**Figure 1 nutrients-17-01051-f001:**
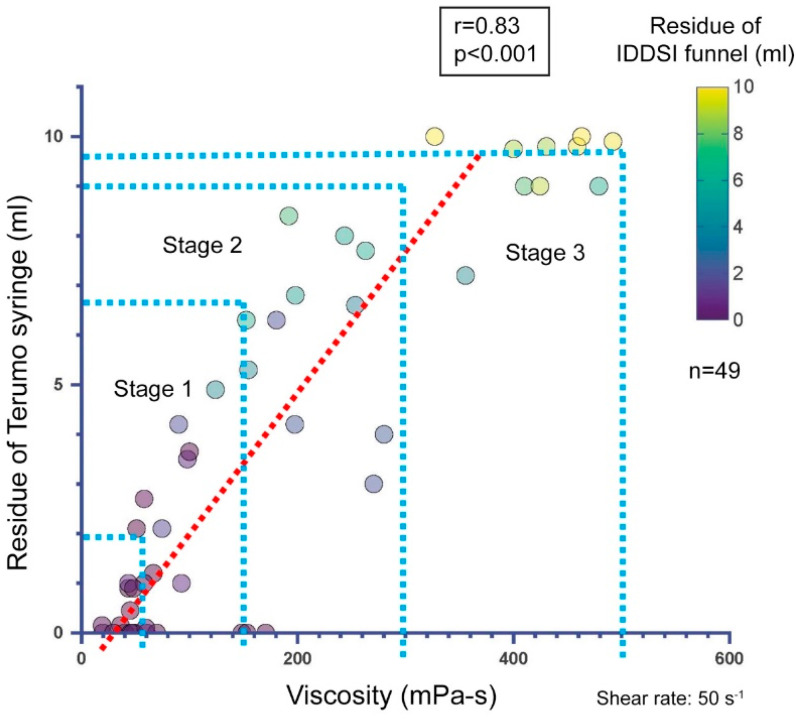
Relationship between viscosity and residue of the Terumo syringe. Viscosity was measured using an E-type viscometer, and the residue was measured with a Terumo syringe (correlation coefficient: r = 0.83, *p* < 0.001, Spearman’s rank correlation analysis). The color in the circle indicates the residual volume of the IDDSI funnel with the same sample.

**Figure 2 nutrients-17-01051-f002:**
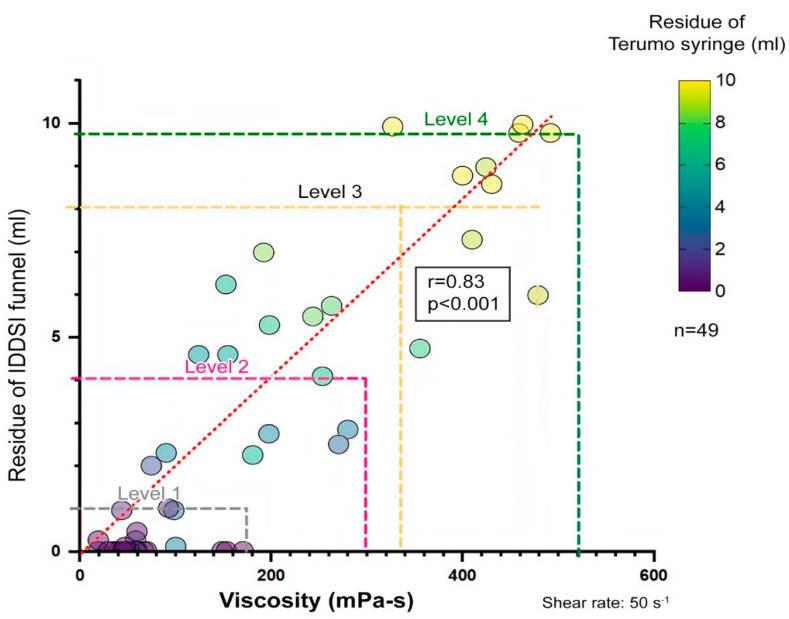
Relationship between viscosity and IDDSI funnel residue. Viscosity was measured using the E-type viscometer, and the residue was measured with the IDDSI funnel (correlation coefficient: r = 0.83 *p* < 0.001, Spearman’s rank correlation analysis). The color in the circle indicates the residual volume of the Terumo syringe with the same sample.

**Figure 3 nutrients-17-01051-f003:**
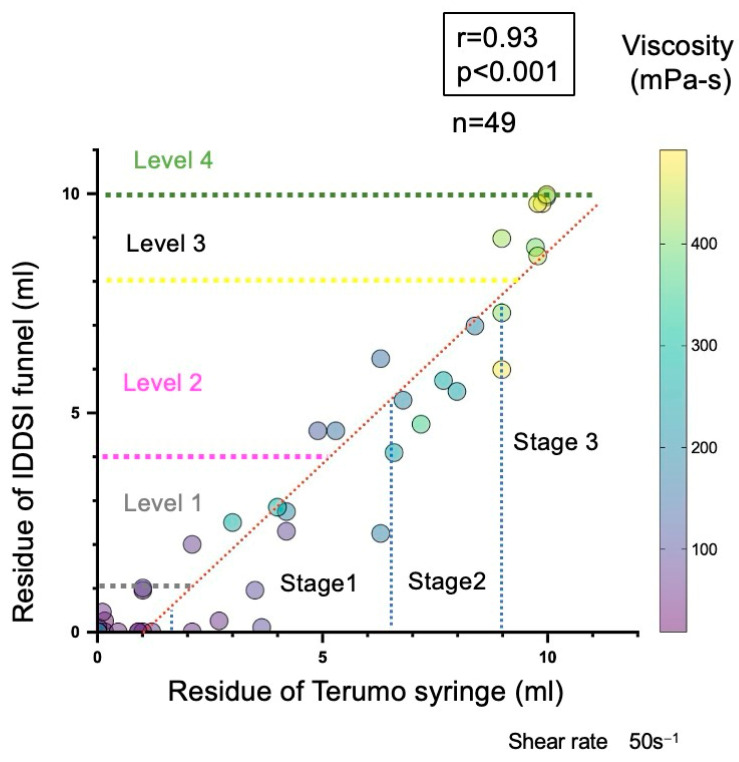
Relationship between residual volumes of the Terumo syringe and IDDSI funnel (correlation coefficient: r = 0.93, *p* < 0.001, Spearman’s rank correlation analysis). The color in the circle indicates the viscosity measured using the E-type viscometer with the same sample. The number beside the circles shows the viscosity of each sample.

**Figure 4 nutrients-17-01051-f004:**
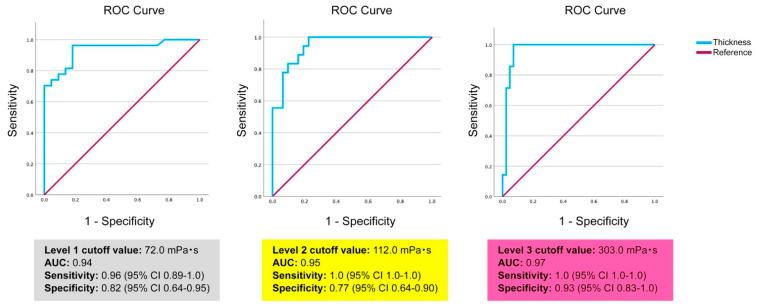
Results of the ROC analysis to determine the cutoff for IDDSI levels 1, 2, and 3 using the viscosity and the date of funnel residue from 49 samples. The cutoffs were determined at maximum sensitivity and specificity.

**Figure 5 nutrients-17-01051-f005:**
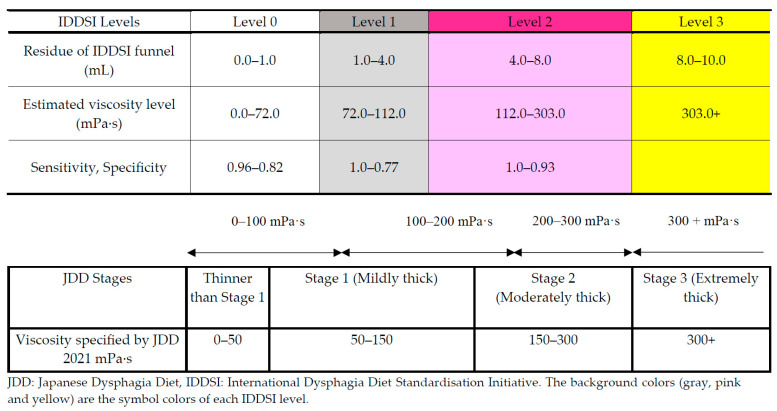
Correspondence between JDD 2021 and IDDSI ver. 2.0 for beverages.

**Table 1 nutrients-17-01051-t001:** Sample beverages, amount of thickening agent, and preparation methods.

No.	Thickening Agent (g) Beverages	Mixing				30 mPa·s	50 mPa·s	100 mPa·s	200 mPa·s	400 mPa·s
		Mixing Time (sec) [1]	Static Time (min)		Mixing Time (sec) [2]					
		30	5	10	30	B	C	D	F	H
1	Mineral water	X				0.3	0.4	0.5	1.0	2.0
2	Tea (Oi Ocha™)	X				0.3	0.4	0.5	1.0	1.6
3	Coffee (Boss™ Black Coffee unsweetened)	X				0.3	0.4	0.5	1.0	1.6
4	Black tea (Afternoon tea™, straight, unsweetened)	X				0.3	0.4	0.5	1.0	1.6
5	Oolong tea	X				0.3	0.4	0.5	1.0	1.8
6	100% Orange juice (Pom Juice)	X	X		X	0.5	0.6	0.7	1.2	1.6
7	Sports drink (Pocari Sweat™)	X	X		X	0.5	0.6	0.7	1.2	1.7
8	Milk (Meiji Oishii Gyunyu™)	X		X	X	0.5	0.6	0.7	1.2	2.0
9	Tomato juice	X	X		X	-	0.5	0.6	0.7	1.2
10	Lactic acid beverage (Yakult™)	X		X	X	0.5	0.6	0.7	1.2	1.8

X: applied.

**Table 2 nutrients-17-01051-t002:** Viscosities of the sample beverages.

**№**	**Target viscosity (mPa·s)**	**30**	**50**	**100**	**200**	**400**
	**Solution n = 2; Average Viscosity (mPa·s)**	**B**	**C**	**D**	**F**	**H**
1	MINERAL water	43.5	58	90	192.5	327.5
2	Tea (Oi Ocha™)	19.5	49	66.5	253.5	400
3	Coffee (Boss™ Black Coffee unsweetened)	36	58	51	180.5	410
4	Black tea (Afternoon tea™, straight, unsweetened)	45.5	48	98	263	430.5
5	Oolong tea (Suntory cooperation)	29.5	45	100	243.5	553.5
6	100% orange juice	40	60	69.5	197.5	355.5
7	Sports Drink (Pocari Sweat™)	47	59.5	92.5	280	481
8	Milk (Meiji Oishii Gyunyu™)	19	43.5	74.5	152.5	479
9	Tomato juice	-	124	154.5	198	424.5
10	lactic acid beverage (Yakult™)	148.5	153	170.5	270.5	463

**Table 3 nutrients-17-01051-t003:** Degree of agreement between the definition of JDD2021 and the results of the JDD syringe test.

Viscosity (mPa·s)	Definition of JDD2021	JDD Syringe Test Using the Terumo Syringe Degree of Agreement with Results
<50	Thinner than Stage 1	92%
50–150	Stage 1 (Mildly thick)	46%
150–300	Stage 2 (Moderately thick)	23%
300–500	Stage 3 (Extremely Thick)	40%

JDD: Japanese Dysphagia Diet.

## Data Availability

The experimental data that support the findings of this study are available in Appendix A.

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
