# Peer review of "Characterization of Beverage Viscosity Based on the International Dysphagia Diet Standardisation Initiative and Its Correspondence to the Japanese Dysphagia Diet 2021"

_nutrients, 2025, doi:10.3390/nu17061051_

Round 1

Reviewer 1 Report

Comments and Suggestions for Authors

Review for Nutrients

Nutrients-3507339

Characterization of Beverage Viscosity Based on the International Dysphagia Diet Standardisation Initiative and its Correspondence to the Japanese Dysphagia Diet 2021

Mari Nakao-Kato 1,2*, Aya Takahashi 1, Jin Magara3

In the study entitled Characterization of Beverage Viscosity Based on the International Dysphagia Diet Standardisation Initiative and its Correspondence to the Japanese Dysphagia Diet 2021, Nakao-Kato et al. designed a study to establish a correspondence between the International Dysphagia Diet Standardisation Initiative (IDDSI) and the Japanese Dysphagia Diet 2021 (JDD2021). To achieve this objective, they analyzed 49 thickened beverage samples using an E-type viscometer, the IDDSI flow test, and the JDD syringe test. The instruments and methodologies employed were well chosen and appropriate for addressing the research question, including the regression models used.

By comparing both classification systems, the findings established that IDDSI Level 0 corresponds to a thinner consistency than JDD Stage 1 (0–72 mPa·s), Level 1 corresponds to Stage 1 (72–112 mPa·s), Level 2 to Stages 1–2 (112–303 mPa·s), and Level 3 to Stage 3 (>303 mPa·s). The authors demonstrated a strong correlation between the two classification systems, reporting an R² value of 0.803. According to the authors, these results “enable healthcare providers to translate between classification systems, improving dysphagia management internationally while supporting evidence-based care and global research.”

Although this manuscript is of interest, several minor revisions need to be addressed before potential publication:

1- The acronym AUD must be defined before its first occurrence in the abstract and throughout the manuscript.

2- References 1 to 6 should be updated with more recent sources. Additionally, references 8 and 10 should be revised, as more recent literature is available on the prevalence and impact of swallowing disorders among U.S. adults and the epidemiology, risk factors, and quality-of-life impact of dysphagia. Other references appear to be current.

3- The Discussion section needs to be rewritten, as it currently reads more like an extension of the Results section rather than a critical analysis of the findings.

4- The Conclusion should be more concise, avoiding detailed results and instead focusing on whether the initial study objectives were achieved.

5- A thorough English language revision is necessary for the entire manuscript to improve clarity, grammar, and readability.

6- The title word Standardisation needs to be correct to Standardization.

Comments on the Quality of English Language

A thorough English language revision is necessary for the entire manuscript to improve clarity, grammar, and readability.

Author Response

Responses to Reviewer 1

Nutrients-3507339

Characterization of Beverage Viscosity Based on the International Dysphagia Diet Standardisation Initiative and its Correspondence to the Japanese Dysphagia Diet 2021

Mari Nakao-Kato 1,2*, Aya Takahashi 1, Jin Magara3

In the study entitled Characterization of Beverage Viscosity Based on the International Dysphagia Diet Standardisation Initiative and its Correspondence to the Japanese Dysphagia Diet 2021, Nakao-Kato et al. designed a study to establish a correspondence between the International Dysphagia Diet Standardisation Initiative (IDDSI) and the Japanese Dysphagia Diet 2021 (JDD2021). To achieve this objective, they analyzed 49 thickened beverage samples using an E-type viscometer, the IDDSI flow test, and the JDD syringe test. The instruments and methodologies employed were well chosen and appropriate for addressing the research question, including the regression models used.

By comparing both classification systems, the findings established that IDDSI Level 0 corresponds to a thinner consistency than JDD Stage 1 (0–72 mPa·s), Level 1 corresponds to Stage 1 (72–112 mPa·s), Level 2 to Stages 1–2 (112–303 mPa·s), and Level 3 to Stage 3 (>303 mPa·s). The authors demonstrated a strong correlation between the two classification systems, reporting an R² value of 0.803. According to the authors, these results “enable healthcare providers to translate between classification systems, improving dysphagia management internationally while supporting evidence-based care and global research.”

Although this manuscript is of interest, several minor revisions need to be addressed before potential publication:

We would like to express our gratitude for your time and thoughtful feedback on our manuscript. In response to your comments, we have made revisions to the manuscript. We will address each of your comments individually.

  1. The acronym AUD must be defined before its first occurrence in the abstract and throughout the manuscript.

Thank you for your comment. We have added the definition of AUC as area under the curve at its first use in the abstract and the manuscript as shown below.

Line 20 (area under the curve [AUC])

Line 269 (area under the curve [AUC])

  1. References 1 to 6 should be updated with more recent sources. Additionally, references 8 and 10 should be revised, as more recent literature is available on the prevalence and impact of swallowing disorders among U.S. adults and the epidemiology, risk factors, and quality-of-life impact of dysphagia. Other references appear to be current.

We appreciate your suggestion and have updated the references as shown below:

Reference #1 Speyer, R.; Cordier, R.; Farneti, D.; Nascimento, W.; Pilz, W.; Verin, E.; Walshe, M.; Woisard, V. White paper by the European Society for Swallowing Disorders: screening and non-instrumental assessment for dysphagia in adults. Dysphagia. 2022, 37, 333-349. doi: 10.1007/s00455-021-10283-7.

Reference #2 Jotz, G.P. Dysphagia: an overview. Int Arch Otorhinolaryngol. 2023, 27, e377-e379.

Reference #3 Yang, S.; Park, J.W.; Min, K.; Lee, Y.S.; Song, Y.J.; Choi, S.H.; Kim, D.Y.; Lee, S.H.; Yang, H.S.; Cha, W.; Kim, J.W.; et al. Clinical practice guidelines for oropharyngeal dysphagia. Ann Rehabil Med. 2023, 47, S1-S26.

Reference #4 Miles, A.; McRae, J.; Clunie, G.; Gillivan-Murphy, P.; Inamoto, Y.; Kalf, H.; Pillay, M.; Pownall, S.; Ratcliffe, P.; Richard, T.; and Robinson, U. An international commentary on dysphagia and dysphonia during the COVID-19 pandemic. Dysphagia 2022, 37, 1349-1374.

Reference #5 Thiyagalingam, S.; Kulinski, A.E.; Thorsteinsdottir, B.; Shindelar, K.L.; Takahashi, P.Y. Dysphagia in older adults. Mayo Clin Proc. 2021, 96, 488-497.

Reference #6 Doan, T.N.; Ho, W.C.; Wang, L.H.; Chang, F.C.; Nhu, N.T.; Chou, L.W. Prevalence and methods for assessment of oropharyngeal dysphagia in older adults: a systematic review and meta-analysis. J Clin Med. 2022, 11, 2605.

Reference #8 Adkins, C.; Takakura, W.; Spiegel, B.M.R.; Lu, M.; Vera-Llonch, M.; Williams, J.; Almario, C.V. Prevalence and characteristics of dysphagia based on a population-based survey. Clin Gastroenterol Hepatol. 2020, 18, 1970-1979.

We have revised the manuscript in accordance with the new article:

Lines 36–37 In the United States, one in six adults reported experiencing difficulty swallowing. However, half of these individuals have not discussed their symptoms with a clinician.

Reference #10 Roberts, H.; Lambert, K.; Walton K. The prevalence of dysphagia in individuals living in residential aged care facilities: a systematic review and meta-analysis. Healthcare. 2024, 12, 649.

Lines 41–42 In a systematic review and meta-analysis, the pooled prevalence of dysphagia in residential aged care facilities was found to be 56.11%.

Reference #11 Mateos-Nozal, J.; Montero-Errasquín, B.; Sánchez García, E.; Romero Rodríguez, E.; Cruz-Jentoft, A.J. High prevalence of oropharyngeal dysphagia in acutely hospitalized patients aged 80 years and older. J Am Med Dir Assoc. 2020, 21, 2008-2011.

Lines 42–43 In a study of acute care hospitals in Spain, the prevalence of hospitalized adults over 80 years old was 82.4%.

  1. The Discussion section needs to be rewritten, as it currently reads more like an extension of the Results section rather than a critical analysis of the findings.

We are grateful for your feedback and have thoroughly revised the discussion. First, the presentation of results has been streamlined, offering a more analytical and critical evaluation of the standard system to date. Constructive considerations and suggestions have been included based on the results of our study. In particular, we have focused more on the viscosity characteristics of the IDDSI levels. We have indicated that the IDDSI framework intentionally avoids the specification of the viscosity range (Lines 312–313), and we have drawn attention to the contradiction that there is a statistically reliable threshold for the transition point of the IDDSI levels based on the results of the receiver operating curve analysis. In addition, the revised section includes a discussion of the practical implications from this study in the management of dysphagia, proposing several ways to apply the results of this study in clinical practice (Lines 338–363). The manuscript also critically examines the current standardization process, showing that the fundamental differences among the various systems are not adequately considered, and proposes the need for an integrated approach for viscosity measurement (Lines 364–372). In addition, based on the results of this study, we suggest potential improvements, including the introduction of correction factors based on beverage characteristics, development of evaluation methods optimized for specific viscosity ranges, development of a flexible framework reflecting cultural diversity and clinical realities, and establishment of a verification system based on clinical outcomes (Lines 373–392).

  1. The Conclusion should be more concise, avoiding detailed results and instead focusing on whether the initial study objectives were achieved.

We appreciate your feedback. We have altered the conclusion to make it more concise and to focus on the outcome of the initial study objectives.

Lines 446–452

This study fully achieved its intended objective of characterizing the relationship between viscosity ranges corresponding to IDDSI levels and the JDD2021 standard. Using multiple measurement techniques, the viscosity characteristics of each IDDSI level were clearly defined and statistical relationships between IDDSI flow test results and absolute viscosity measurements were established. Our research will contribute to the development of international clinical practices, research methods, and standardization efforts in dysphagia management.

5- A thorough English language revision is necessary for the entire manuscript to improve clarity, grammar, and readability.

We have reviewed the manuscript with the aim of improving clarity, grammar, and readability. (The words and sentences with blue highlight are changed for the English language revision.)

6- The title word Standardisation needs to be correct to Standardization.

Thank you for your comment. As "The International Dysphagia Diet Standardisation Initiative" is a proper noun (https://www.iddsi.org/about-us/overview), we have retained the spelling used by the organization.

Reviewer 2 Report

Comments and Suggestions for Authors

The English text is scientifically accurate, and the study represents a significant advancement in the standardization of viscosity measures related to dysphagia management. It provides a detailed analysis of the relationship between the IDDSI and JDD2021 classifications, supporting its conclusions with ROC analysis and multivariate regression. In future research, it would be beneficial to include a larger sample size to obtain more precise results, as the small sample size may introduce bias.

My question is as follows: To what extent do the findings of this study assist practicing clinicians in dysphagia management, and how could the standardization process be further improved? What additional research would be necessary to ensure that viscosity standardization is truly unified on an international level?

The plagiarism index of the article is low, the English language is appropriate and well-understood, the statistical analysis is thorough and accurate, and the references are relevant. Additionally, it would be helpful to include an abbreviation list below the tables e g JDD IDDSI etc.

Author Response

Responses to Reviewer 2

Comments and Suggestions for Authors

The English text is scientifically accurate, and the study represents a significant advancement in the standardization of viscosity measures related to dysphagia management. It provides a detailed analysis of the relationship between the IDDSI and JDD2021 classifications, supporting its conclusions with ROC analysis and multivariate regression.

In future research, it would be beneficial to include a larger sample size to obtain more precise results, as the small sample size may introduce bias.

We are grateful for your objective and positive evaluation of this study. We have carefully considered your comments regarding the bias introduced by the sample size and have included this in the limitations section of the manuscript. In future research, we will take your comments into account and aim to verify the results in a larger sample size.

Please find below our responses to your individual questions.

Question1 and 2 :

My question is as follows: To what extent do the findings of this study assist practicing clinicians in dysphagia management, and how could the standardization process be further improved?

To answer these questions, the Discussion section has been expanded. Two new subsections have been added: 4.3 Practical impact on dysphagia management in clinics or facilities and 4.4 Potential for improvement in standardization processes.

In section 4.3 (Practical impact on dysphagia management in clinics or facilities), we highlight four points that may lead to an improvement (Lines 341–363).

  1. Refinement of clinical assessment and precision: Clinicians will be able to accurately convert between both classification systems, facilitating smooth communication.
  2. Individualized approaches based on beverage characteristics provide a scientific basis for adjusting thickened beverages, which can contribute to optimal nutritional intake and safety in clinical practice (Lines 347–352).
  3. Enhanced risk assessment requires clinicians to be more cautious in their assessment and interpretation of the viscosity range where the accuracy of flow measurements is low (Lines 353–358).
  4. Streamlined interfacility and international collaboration allows information sharing and continuous care between facilities that use different classification systems (Lines 359–363).

In section 4.4 (Potential for improvement in standardization processes), we have discussed the following points (Lines 364–392):

The current limitations of the standardization process have been mentioned, along with reference to the fact that the IDDSI framework deliberately avoids specifying viscosity ranges (Lines 312–313) and the contradiction that the results of ROC analysis show that there is a statistically reliable threshold for the transition points of IDDSI levels. The need for an integrated approach for viscosity measurement has also been mentioned. The multiple regression analysis in this study showed that beverage characteristics exert an influence on the final physical properties of the beverage; similarly, the final properties of the beverage affect its physical properties. It is also recommended that correction factors based on beverage properties be incorporated into the standardization process. The present study demonstrated that the accuracy of measurements obtained using syringes and funnels may vary depending on the viscosity range. This suggests that reliability can be enhanced by introducing specific evaluation methods for different viscosity ranges. A significant challenge in the realm of international standardization pertains to the delicate balance between regional adaptation and the necessity for a flexible framework. In addition to an explicit definition of viscosity values for accurate comparison, it is imperative to consider the development of supplementary guidelines for ingredients that are specific to certain regions. Furthermore, it is crucial to verify whether the classification system contributes to tangible clinical outcomes.

Question 3; What additional research would be necessary to ensure that viscosity standardization is truly unified on an international level?

We have discussed this in a newly created subsection in the Discussion section (4.6 Additional Research Needed for International Standardization, lines 418–444). This section covers the necessity of collaborative verification of measurements across multiple countries, systematic comparisons using different types of thickening agents, prospective studies using patient-centered outcomes, and analysis of the impact of food culture and eating habits.

The plagiarism index of the article is low, the English language is appropriate and well-understood, the statistical analysis is thorough and accurate, and the references are relevant.

Thank you for your positive feedback.

Additionally, it would be helpful to include an abbreviation list below the tables e g JDD IDDSI etc.

Thank you for your suggestion. We have included an explanation of the abbreviations used below the relevant tables.
